# Synthesis and characterization of attosecond light vortices in the extreme ultraviolet

R. Géneaux[1], A. Camper[2], T. Auguste[1], O. Gobert[1], J. Caillat[3], R. Taïeb[3] & T. Ruchon[1]

Infrared and visible light beams carrying orbital angular momentum (OAM) are currently thoroughly studied for their extremely broad applicative prospects, among which are quantum information, micromachining and diagnostic tools. Here we extend these prospects, presenting a comprehensive study for the synthesis and full characterization of optical vortices carrying OAM in the extreme ultraviolet (XUV) domain. We confirm the upconversion rules of a femtosecond infrared helically phased beam into its high-order harmonics, showing that each harmonic order carries the total number of OAM units absorbed in the process up to very high orders (57). This allows us to synthesize and characterize helically shaped XUV trains of attosecond pulses. To demonstrate a typical use of these new XUV light beams, we show our ability to generate and control, through photoionization, attosecond electron beams carrying OAM. These breakthroughs pave the route for the study of a series of fundamental phenomena and the development of new ultrafast diagnosis tools using either photonic or electronic vortices.

[1] LIDYL, CEA, CNRS, Université Paris-Saclay, CEA Saclay, 91191 Gif-sur-Yvette, France. [2] Department of Physics, Ohio State University, Columbus, Ohio 43210, USA. [3] Sorbonne Universités, Laboratoire de Chimie Physique—Matière et Rayonnement (UMR7614), UPMC Univ Paris 06, 11 rue Pierre et Marie Curie, 75005 Paris, France. Correspondence and requests for materials should be addressed to T.R. (email: thierry.ruchon@cea.fr).

Like massive particles may carry two types of angular momenta, namely spin and orbital angular momenta (SAM and OAM, respectively), the massless photon can be assigned two such characteristics[1]. It was recognized earlier that SAM is associated with the circular polarization of light beams, but only 20 years ago was the OAM of light associated with beams with a tilted wavefront[2]. The unique properties of twisted light beams and their large availability in the infrared and visible spectral regions lead to the emergence of countless applications, from quantum information[3] to microscopy[4], nanoparticle manipulation[5] or fine structuring of materials using pulsed lasers[6]. In the extreme ultraviolet (XUV) spectral range, techno-logical applications using OAM beams were anticipated[7,8], building on predictions of specific light matter interactions[9,10]. However, because of the lack of sources, most of these predictions could not be confronted to the experiment. Here we first demonstrate a route based on high harmonic generation to synthesize and characterize helically shaped XUV trains of attosecond pulses. Using this photon source, we also report the experimental synthesis of electronic vortices with attosecond time structure. These breakthroughs pave the route for the study of a series of fundamental phenomena and the development of new ultrafast diagnosis tools using either photonic or electronic vortices or springs[11,12], for instance, the observation of new kinds of dichroisms[7,8] or the visualization of dislocation strain fields in bulk material[13].

The most common implementation of macroscopic beams carrying OAM, typically produced using spiral staircase phase plates, displays a helical phase front, with a phase singularity associated with zero intensity along the axis[5]. At the photon level, the twisted wavefront of the beam translates into a quantized momentum, with its component along the beam axis taking discrete $\ell\hbar$ values, where $\ell$ is a positive or negative integer. Laguerre-Gaussian modes (LG) are eigensolutions of the paraxial wave equation and form a natural set for these helically phased beams. Lately, motivated by theoretical predictions, the development of sources of XUV beams carrying OAM was undertaken on both quasi-continuous synchrotron installations[14], and on femtosecond free electron lasers[15,16]. High-harmonic generation (HHG)-based XUV sources constitute a table top, ultrashort and largely tunable alternative to those large-scale instruments. They show unrivalled stability specifications, especially useful for pump–probe experiments targeting attosecond temporal resolution[17]. They are based on the upconversion of a high-intensity femtosecond visible–infrared laser beam into the XUV range through highly nonlinear interaction with a gas. When driving HHG with helically phased light beams, momentum conservation predicts a 'multiplicative' rule for OAM transfer[11], similar to what is known in low-order nonlinear processes[5]. The OAM of the harmonic order $q$ is then given by the quantum number $\ell_q = q\ell_1$, where $\ell_1$ is the OAM of the driving field. Two groups measured the helicity of such a harmonic beam: first, Zürch et al.[18] unexpectedly observed, on a single harmonic, that $\ell_q = \ell_1$. It was argued that this disagreement with theory, which was suggesting violation of momentum conservation, was due to parametric instabilities preventing higher-order vortices from propagating towards the detector[19]. Later, Gariepy et al.[20] succeeded, through fine interferometric measurements performed on three harmonics, in demonstrating the expected multiplicative rule. Therefore, OAM–HHG holds the potential for producing XUV pulses of attosecond duration carrying OAM. The shape of such a broad comb of phase-locked helically phased harmonics has been theoretically described[11]. Such attosecond light springs, as they were dubbed[12], could in turn be a unique source to tailor attosecond electron springs through photoionization. Currently,

twisted electron beams, primarily generated through tailoring Gaussian electron beams in the quasi-static regime[21,22], are actively studied for future applications in fields covering spectroscopy of diluted and condensed matter[23], microscopy and particle physics[24].

In the present work, we report on the synthesis of attosecond XUV 'light springs'. First, we show that the multiplicative OAM transfer rule for HHG is valid in our experimental conditions—without relying on any diffractive or interferometric technique. Importantly, this approach is directly scalable to an arbitrary spectral range. We then present the results of a two-colour ionization experiment in which we are able to characterize the attosecond structure of the generated light spring, which is in turn used to generate attosecond helically shaped electron beams.

## Results

**Link between OAM and divergence in HHG.** Twisted attosecond XUV pulses hold great promises for a variety of applications. However, as raised up in ref. 20, a critical point is the characterization of OAM on a broad wavelength range. For instance, the study of helical dichroism in solids is envisioned close to transition-metal K-edges[7], which lie at very short wavelengths. In much the same way, the extremely short time resolutions attainable using HHG-based sources are only reached when using a broad spectrum covering a large number of harmonics. In the visible–infrared range, the characterization is usually done directly using diffractive optical elements or interference schemes[5]. These approaches were transferred to the XUV domain in the aforementioned lines of work[18,20], requiring the resolution of fringes with extremely low spacing because of the short wavelength. These procedures clearly get ever more difficult to implement as the wavelength gets shorter. Here, for our broad harmonic comb, we use a different approach to measure the OAM of the harmonic field, which does not rely on interferometric measurements. It is based on another signature of the OAM value: the ring-shaped intensity radial profiles associated with the vortices' phase singularity. In particular, it can be shown that the multiplicative law, $\ell_q = q \times \ell_1$, is the only one that yields a single ring with a constant diameter proportional to $\sqrt{|\ell_1|}$ over the whole spectrum. Indeed, unlike what is stated in the Supplementary Information of ref. 18, the enlargement of the ring's diameter because of the increase in the OAM is exactly compensated for by the smaller diffraction observed at shorter wavelengths. This statement is supported by both a simple analytical model (see Supplementary Notes 1 and 2) and the results of numerical simulations of HHG[25] illustrated in Fig. 1. The simulations were carried out in a similar way as in ref. 11 but taking into account all aspects of a real experiment, including the propagation of the infrared beam towards the gas target (see Methods for more details). As shown in Fig. 1a, the harmonics at focus have a ring-shaped intensity as well as a spiral-shaped phase running $q$ times $2\pi$ along the ring. When propagated to the far field, the harmonic profiles show one intense central ring (insets of Fig. 1b), suggesting that the generation is dominated by the emission of a single LG mode. Note that extra faint rings appear around it. They result from both the nonlinearity of the process and an interplay between two different contributions to HHG, associated with distinct electron quantum trajectories at the single atom level (so-called 'short' and 'long'), as explained in ref. 26.

The diameter of the harmonic rings for two different values of $\ell_1$ is plotted in Fig. 1b. As expected, the diameter is mostly constant apart from a jump around H25, which is precisely where the 'short' and 'long' trajectory contributions are merging. This leads to interferences and a modulation of the spatial profile of the harmonics, similarly to what is observed with Gaussian

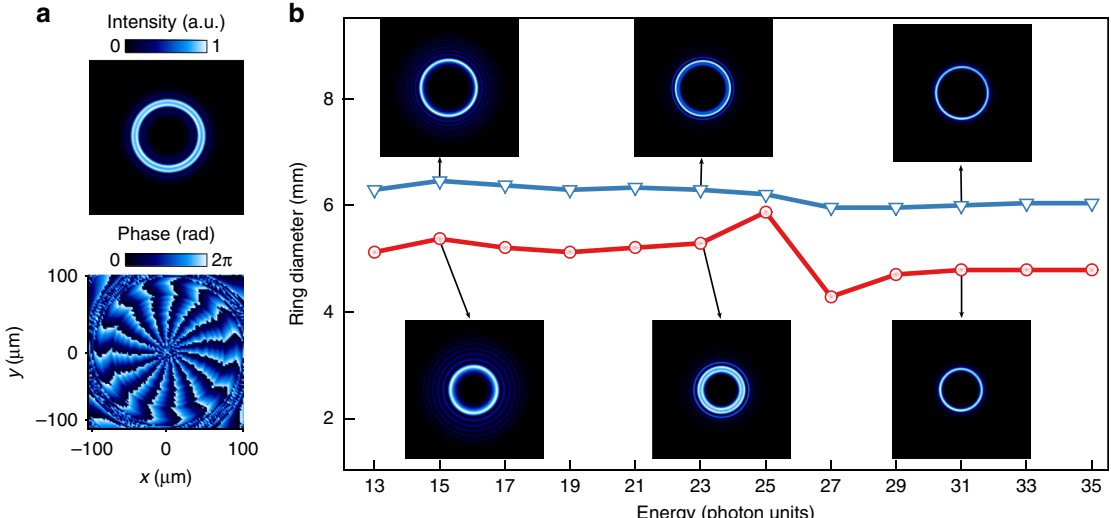

**Figure 1 | Calculation of high-order harmonic profiles generated with a driving laser carrying OAM.** (**a**) Intensity and phase transverse profiles of harmonic 15 close to focus when the driving laser carries $\ell_1 = 1$. We observe a well-defined ring-shaped distribution for the intensity (top panel) and a phase profile linearly increasing by $15 \times \ell_1 \times 2\pi$ along a circle centred on the beam axis (bottom panel). (**b**) Diameter of the harmonic rings in the far field for $\ell_1 = 1$ (red circles) and $\ell_1 = 2$ (blue triangles), which present a constant diameter proportional to $\sqrt{|\ell_1|}$. Insets: corresponding intensity profiles for a selection of harmonics.

driving beams[27]. The scaling of the divergence is however very different from the usual Gaussian case, for which the harmonic divergence noticeably changes with harmonic order[28]. When doubling the OAM carried by the driving field $\ell_1$, the harmonic diameter increases by a factor of $1.4 \simeq \sqrt{2}$, consistent with the predicted $\sqrt{|\ell_1|}$ scaling law.

**Generation of attosecond light vortices.** The numerical results reported in Fig. 1 were tested experimentally on the LUCA laser facility at CEA Saclay using the experimental set-up described in Fig. 2a. A spiral phase plate (SPP) is inserted to imprint a spiral staircase phase profile on this incoming beam, which is either one, two or three times $2\pi$ per round. Focusing such a phase-shaped beam leads to a distribution of light about the focal point very close to a LG mode carrying, respectively, $\ell_1 = \pm 1$, $\pm 2$ or $\pm 3$ units of OAM, the sign being determined by the orientation of the phase mask[29]. This is illustrated by the intensity profile reported in Fig. 2b and accurately measured in Supplementary Note 3. XUV harmonic spectra obtained using an infrared driving field with $\ell_1 = 1$, 2 or 3 are displayed in Fig. 2c. For each harmonic order and all OAM values of $\ell_1$, we observe a clear ring shape pinched in the horizontal dimension by the dispersion of the grating. The divergence of the harmonics, seen in the vertical dimension, appears to be constant throughout the whole spectrum, as predicted.

We measured the average ring diameters to be $1.01 \pm 0.02$, $1.33 \pm 0.04$ and $1.61 \pm 0.01$ mm for $\ell_1 = 1$, 2 and 3, respectively, in agreement within 5% with the anticipated $\sqrt{|\ell_1|}$ dependency. We note that we observe only one ring. This is probably because of the selection through phase matching of only the short quantum trajectory. Interestingly, we also note that the harmonics are mostly constituted of one single LG mode. The evidence here is that they show a single ring pattern both at focus (where their shape necessarily mimics that of the driving infrared) and in the far field, where they were measured. The diameter of the rings being constant throughout the spectrum, the momentum is conserved and the OAM is multiplied by the harmonic order. Here the highest value is obtained when using $\ell_1 = 3$, for which $\ell_{19} = 57$. We also carried out this experiment in Neon, allowing

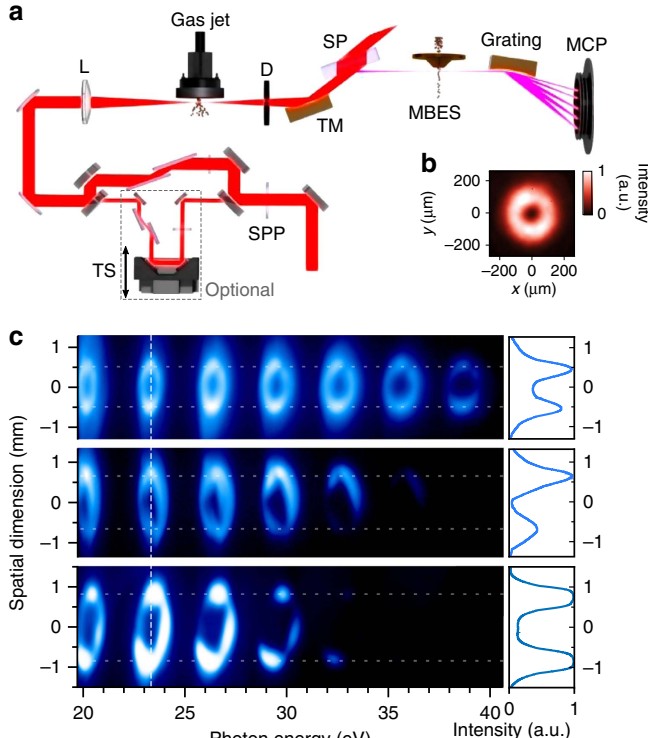

**Figure 2 | Experimental observation of HHG spectra carrying OAM.** (**a**) Experimental set-up. It has two operation modes, corresponding to the imaging of HHG light spectra and to the characterization of EWP. The close to Gaussian Ti:Sapphire laser beam is passed through a SPP and used to generate harmonics. They can be either directly imaged using a photon spectrometer or focused inside a MBES, where an extra infrared beam can be added with a delay controlled by a translation stage (TS), allowing to perform RABBIT measurements (Methods). (**b**) Intensity profile of the laser beam at focus close to the HHG gas inlet. (**c**) Normalized intensity of harmonic orders 15th to 27th generated in argon and observed in the far field, using $\ell_1 = 1$ (top row), $\ell_1 = 2$ (middle row) and $\ell_1 = 3$ (bottom row).

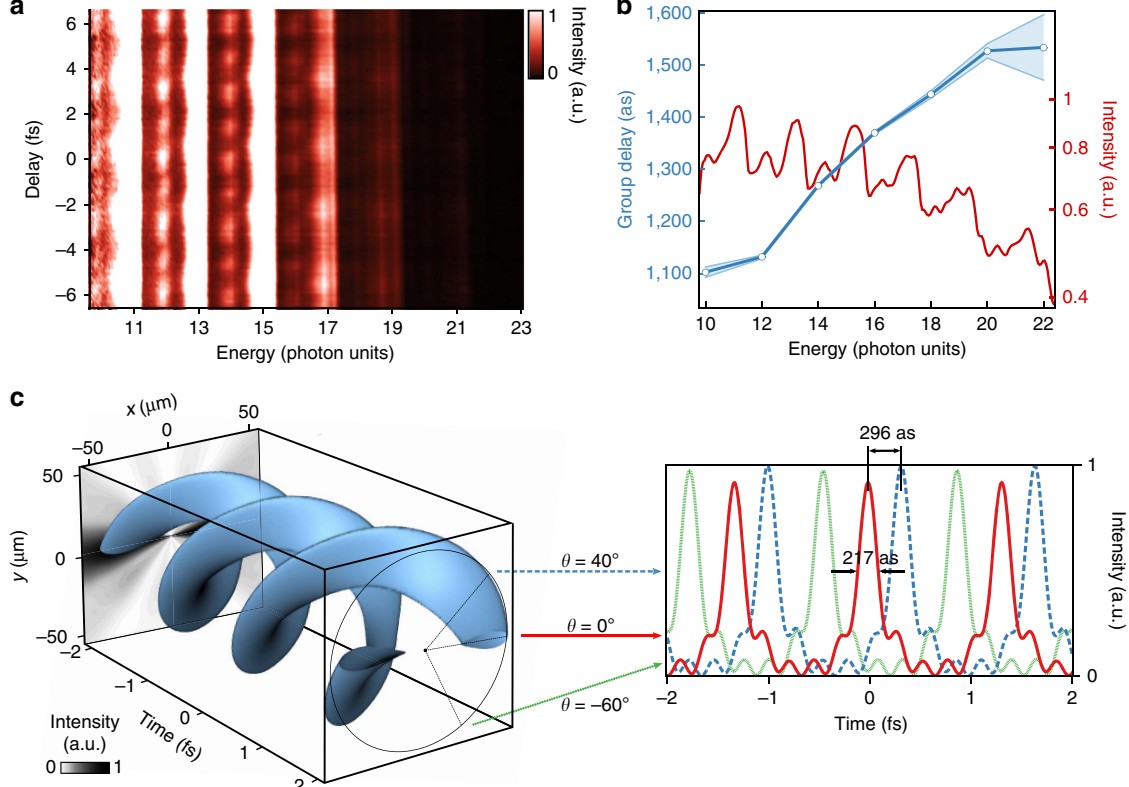

**Figure 3 | Attosecond electronic beams carrying OAM.** (**a**) Two-colour XUV + infrared two-photon photoionization spectrogram of argon for a driving field with $\ell_1 = 1$. The main lines correspond to odd harmonics, while the weaker oscillating ones, showing a double periodicity of $T = 1.33$ and 2.7 fs, are SBs. The collection efficiency of our spectrometer limits the number of measurable SBs even if the harmonic cutoff lies higher. (**b**) Delay-averaged spectrum in log scale (red) and group delay of the photoionized EWP (blue circles, the light blue strip about the curve represents the numerical analysis error bar at $3\sigma$). (**c**) Spatiotemporal shape of the photoelectrons emitted in the forward direction using the intensity profiles from Fig. 2c, the group delays from **b**, and assuming a flat cross-section for argon in this energy range. The 3D surface is a contour at 80% of the maximum intensity. The black and white back panel is a projection of this intensity in the $t = -2$ fs plane. (**d**) Temporal cuts at three different azimuthal angles in the electron beam $\theta = 0°$ (red), 40° (dashed blue) and $-60°$ (green).

us to reach an OAM $\ell_{41} = 41$ with $\ell_1 = 1$ while keeping a clean spatial profile (see Supplementary Fig. 3). Combined with the spectral phase locking inherent to the HHG process, this opens a way of synthesizing attosecond light pulses carrying extremely high mean values of OAM. In the next section, we demonstrate that the harmonics generated with a helically phased infrared beam are phase-locked and use the resulting attosecond 'light spring' to shape electron wave packets (EWP) through photoionization.

**Attosecond electron vortices.** Since the early days of attophysics it has been recognized that attosecond XUV pulses could tailor EWP through photoionization[17,30]. Applications of such attosecond EWP were proposed, for instance, as a quantum stroboscope[31], or as tools to localize EWP in space and time around atoms and molecules[32]. In the present context, we propose to increase the tailoring knobs by transferring the OAM carried by the harmonics on photoionized electrons, resulting into electron bursts carrying OAM. We demonstrate the synthesis of such helically shaped electron beams through an interferometric measurement based on the well-established XUV–infrared cross-correlation technique called RABBIT[33] (reconstruction of attosecond beating by interference of two-photon transitions), in which synchronized XUV harmonics and infrared beams ionize a target gas. Thus, an electron can be promoted to a given final state called sideband (SB) through two quantum paths involving the absorption of one photon coming

from consecutive harmonic orders and the absorption/emission of one infrared photon. The resulting quantum interference leads to oscillations of the SB yield as a function of the XUV/infrared delay at twice the infrared laser period and its phase is the sum of the total relative phase of the harmonic orders involved plus that of the two-photon dipole transitions at play, denoted by $\Delta\varphi_q$.

Most importantly, for the interference to be observed, a constant phase relation between the XUV and infrared fields across the gas jet is required. Using a twisted XUV beam (see Supplementary Note 4), this condition is met with a dressing infrared beam carrying the OAM $\ell_1$, but not with a standard Gaussian dressing beam (see Methods).

Our experimental test is based on a Mach–Zehnder interferometer sketched in Fig. 2a. Scanning the XUV–infrared delay, we observe the expected $2\omega$ oscillations of the SBs' intensity (Fig. 3a). We also see some $\omega$ oscillations, which are due to a modulation of HHG by the second infrared field but do not hinder the $2\omega$ SB analysis[33]. As in standard RABBIT measurements, the phase of the $2\omega$ oscillation is directly linked to the group delay (GD $\simeq \Delta\varphi_q/2\omega$, with $\omega = 2\pi/\lambda$) of the EWP generated at the atomic level by the harmonic comb. This GD exhibits a linear dependence, with an increase of $\Delta t_e = 103 \pm 9$ as between consecutive odd harmonic orders, which is consistent with the reported values using a Gaussian beam with the same driving wavelength and peak intensity[33]. The presence of the $2\omega$ oscillations demonstrates that the azimuthally spinning infrared intensity is matched with the XUV light spring on the subcycle

timescale. As explained in the Methods section, this is again consistent with momentum conservation. Supplementary Fig. 5 shows that, if we dress the harmonics with a Gaussian infrared beam, the oscillations wash out as expected. The RABBIT trace also demonstrates the spectral phase-locking of the high harmonic orders, thus confirming the attosecond 'light spring' structure of the XUV bursts generated with a helically phased infrared beam.

## Discussion

The two sets of measurements reported above, that is, the spatial intensity profile of the harmonics and their GD finally provides us with all data needed to reconstruct the full spatiotemporal shape of the emitted attosecond electron beam. We hereafter focus in the case of photoelectrons with linear momentum along the propagation axis, which directly reproduce the spring structure of the XUV beam. As lately predicted[11,12], owing to the low GD over our spectral range, we get two intertwined spring-like structures (Fig. 3c). The helical shape is a direct consequence of the ionizing XUV light structure, while the presence of two spirals is reminiscent of the generation of odd harmonics only. A cut in the spatial domain displays a double lobe structure, while the temporal profile shows a train of attosecond pulses lasting ∼ 200 as (Fig. 3d). An interesting characteristic of this structure is that it leads to an identical attosecond pulse train when observed at different azimuthal positions, only delayed by 7.4 as.degree$^{-1}$ (half a period of the infrared driving laser (1,330 as) in 180 degrees). Turning back to light springs, as shown in ref. 11, the corkscrew structure still holds in the case of single attosecond pulses. We believe that this property makes such pulses powerful tools for transient absorption spectroscopy experiments, which require tunable pump–probe delays on the attosecond timescale. Here one may angularly map the attosecond time delay in a single shot, getting rid of any stability requirement. The dynamical range for the delay scan is here 1.33 fs, that is, the infrared laser half optical cycle, which could easily be increased up to several femtoseconds by increasing the driving field wavelength.

In conclusion, we reported on the synthesis and full characterization of attosecond XUV pulses and electron beams, both carrying OAM. The experimental evidence for the transfer of OAM is obtained over an extremely broad spectral range down into the XUV region. This divergence-based analysis method can easily be generalized to arbitrarily large spectral bandwidths. The attosecond structure of the generated XUV pulse train was determined through a photoionization-based cross-correlation technique. First, it confirms the conservation of the standard photoionization selection rules when using helically phased XUV light beams. Second, it opens the route to the manipulation of attosecond electron beams carrying OAM, which will use the large panel of attosecond tools developed in the past 15 years, may it be through high harmonics spectroscopy or XUV–infrared pump–probe experiments. In particular, helical dichroisms were predicted in the XUV spectral range and still remain to be observed. Multicolour HHG using such tailored beams could also provide all-in-one pump–probe schemes, taking advantage of the spatial encoding of the delay in the azimuthal coordinate of these beams. Finally, the accurate characterization of these new light beams will pave the way to the study of the controversial coupling between SAM and OAM in matter during photoionization.

## Methods

**Theoretical details.** In this paragraph we provide details about the simulation of HHG using a driving laser carrying an OAM. For the infrared beam, we accounted for the full experimental set-up sketched in Fig. 2a. To this end, the beam was propagated through the different optical elements, by means of the Huygens–Fresnel integral, up to the gas jet entrance. We consider a spatially and

temporally incident Gaussian beam of 6.25 mm waist and 50 fs duration (full width at half maximum, FWHM). It is propagated through a SPP and focused by a 1 m focal length lens, in the middle of a 500 μm wide (FWHM) Lorentzian argon jet. The wavelength $\lambda$ is 800 nm and the phase plate here imposes a $\ell_1 \times 2\pi$ phase shift along a circle centred on the beam axis ($\ell_1 = 1$ here). The equivalent Gaussian beam waist at focus $w_0$ is 40 μm, leading to an equivalent Rayleigh range $z_R$ of 6.5 mm. The maximum gas pressure is 10 mbar and the laser peak intensity at focus is $1.5 \times 10^{14}\,\mathrm{W\,cm^{-2}}$. This is the starting point for HHG calculations. The structure of the LG mode, which does not possess the cylindrical symmetry of a Gaussian beam, required performing four-dimensional (three-dimensional (3D) in space + time) simulations. In brief, the coupled propagation equations for the driving laser and harmonic fields are numerically solved on a 3D spatial grid, in the paraxial and slowly varying envelope approximations, using a standard finite-difference method. Calculations are performed in a frame moving at the group velocity of the laser pulse, that is, the speed of light in vacuum here. Both electron and atomic dispersions are taken into account. We first computed the driving field at a given time $t$ in the pulse envelope and position $z$ along the propagation axis. This field is then used for calculating the space- and time-dependent ionization yields and dipole strengths. Ionization rates are modelled using Ammosov–Delone–Kraïnov tunneling formula[34], while dipoles are computed in the strong field approximation, following the model described in ref. 35. Once these two terms are obtained, the equation of propagation for the harmonic field is solved. The calculation is repeated for each $t$ and $z$. The SPP inducing the OAM is modelled by $\varphi(x,y) = \ell_1 \tan^{-1}\frac{y}{x}$, where $x$ and $y$ are the coordinates in the plane perpendicular to the propagation axis. The results of the calculations are reported in Fig. 1. As expected, HHG occurs along the peak intensity ring of the generating infrared beam. The FWHM of the profile obtained by a cut along a radius of this ring (hereafter called the thickness of the ring) is a fraction of the thickness of the ring of the driving field, a consequence of the high nonlinearity of HHG. As for the spatial phase, a typical example is given in Fig. 1 by the 15$^{th}$ harmonic order (H15), showing a spiral running 15 times $2\pi$ along the ring. This implies that the H15 photons carry 15 times the OAM of the fundamental frequency photons. This behaviour is consistent with the multiplicative law of perturbative nonlinear optics, and was observed here for all computed harmonic orders, from H11 to H33. Finally, to mimic the experiment, we simulate the propagation of the XUV beam towards an observation plane located 80 cm downstream the gas cell. When reaching the detector the harmonic beam still displays a ring shape, suggesting that generation at focus is dominated by the emission of a single LG mode. The ring diameter increases by a factor of $1.4 \simeq \sqrt{2}$ when doubling the helicity of the phase. This may be guessed from analytical considerations presented in Supplementary Note 1, which show, under reasonable hypotheses, that the divergence of harmonics goes like $\sqrt{|\ell_1|}$.

**Experimental details.** The experimental set-up is sketched in Fig. 2a. The experiments were performed using the LUCA laser server at CEA Saclay, which delivers 30 mJ, 50 fs, 800 nm pulses at 20 Hz. The laser Transverse Electromagnetic Mode TEM$_{00}$ was converted using a 16-level SPP manufactured by SILIOS Technologies. For RABBIT measurements, the laser is split into two uneven parts using a mirror with a 8 mm hole. The main (outer) part of the beam is focused by a 1 m focal length lens (L) into a gas jet provided by a piezoelectric driven valve (Attotech). We could verify that even after reflection off a drilled mirror, the infrared focus kept a donut shape and so did the harmonics, with again a constant divergence. A diaphragm (D) removes the remaining infrared beam, while the harmonics are focused by a 0.5 m focal length toroidal mirror into the sensitive region of a 1 m long magnetic bottle electron spectrometer time of flight (MBES). A SiO$_2$ plate serves as further attenuation of the infrared beam. An extra plain weak infrared beam (4 mm diameter, energy of 70 μJ) may be superimposed and synchronized with the XUV beam in the MBES with a delay controlled by a piezoelectric transducer. Focus imaging in the sensitive region of the electron spectrometer reveals a thick donut profile for the dressing beam, ensuring a homogeneous dressing of the XUV. For intensity measurements, the drilled mirrors are replaced by plain mirrors and the lens has a 2 m focal length. The harmonics are collected downstream the MBES on a photon spectrometer made of a variable line spacing Hitachi grating (001-0437) and a micro channel plate coupled to a phosphor screen. In order to reveal the 2D spatial profile of the harmonics while spectrally resolving them, the MCP are placed 8 cm before the flat field of the grating. The phosphor screen is imaged with a Basler A102f CCD camera. The observation distance from the source to the MCP is 115 cm. We could generate harmonics in argon gas with $\ell_1 = 1$, 2 and 3 (Fig. 1) and neon gas with $\ell_q = 1$ up to H41 (Supplementary Fig. 3). In all cases, we observed a constant ring diameter over the whole XUV spectral range, confirming the general validity of our first measurements in argon. The maximum value of OAM obtained was $\ell_{19} = 57$ using $\ell_1 = 3$ in argon.

**RABBIT with a beam carrying an OAM.** The theory of quantum interferometry used to characterize attosecond XUV pulses usually consider flat wavefronts for both the XUV and dressing infrared beam. Here we anticipate the wavefront of the XUV to be tilted. To be more specific, we assume that the spatial phase of the $q$-th harmonic within the beam is $\Phi_q(R,\theta) = \ell_q\theta$, where the $(R,\theta)$ polar coordinates refer to the beam axis ($R = 0$). The measured SB intensity is an average of the

contributions of all emitted electrons in the interaction region. If we write the intensity coming from one point in this region, neglecting the contributions of the so-called atomic phase, we get for the $2\omega$ oscillating component[33]:

$$SB_{q+1}(\omega, R, \theta) = \cos\left[2\omega\tau_0 + \varphi_{q+2} - \varphi_q + \Phi_{q+2}(R,\theta) - \Phi_q(R,\theta) - 2\Phi_{IR}(R,\theta)\right]$$
$$= \cos\left[2\omega\tau_0 + \varphi_{q+2} - \varphi_q + \left(\ell_{q+2} - \ell_q - 2\ell_1\right)\theta\right],$$

where $\omega$ is the angular frequency of the driving laser and $\varphi_q$ the spectral phase of the $q$-th harmonic order. For the intensity to keep oscillating after integration over $\theta$, which is the operation mode of our detector, the $\theta$-dependent term must vanish, giving $\ell_{q+2} - \ell_q = 2\ell_1$. In particular, if dressing with $\ell_1 = 1$, this condition is only met when having $\ell_q = q\ell_1$. The observation of SBs in this case is therefore another measurement of the multiplicative rule for OAM transfer. Supplementary Fig. 5 shows that the $2\omega$ oscillation component disappears when using $\ell_1 = 0$, for which the $\theta$-dependent term does not cancel. The RABBIT[33] analysis of the SBs' oscillations was carried out under the assumption of fully coherent light[36].

**Data availability.** The data that support the findings of this study are available from the corresponding author (T.R.) upon request.

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

## Acknowledgements

We are particularly grateful to Vincent Gruson, Pascal Salières, Fabien Quéré and Bertrand Carré for stimulating discussions and fruitful suggestions. T.R. acknowledges Antonio Zelaquett Khoury (Univ. Fed. Fluminense, Brazil) for inviting him and introducing him to this topic. This work was supported by the French Agence Nationale de la Recherche (ANR) through XSTASE project (ANR-14-CE32-0010). A.C. acknowledges support of the US Department of Energy, Office of Science, Office of Basic Energy Sciences under contract DE-FG02-04ER15614.

## Author contributions

R.G., A.C. and T.R. conceived, built and carried out the experiment and analysed the data. O.G. developed the laser system and the mode-filtering stage. T.A. did the HHG simulation. J.C. and R.T. did the RABBIT simulations. All authors contributed to the writing of the manuscript.

## Additional information

**Competing financial interests:** The authors declare no competing financial interests.

