## [Peer Review File · Nature Communications]

Reviewer #1 (Remarks to the Author):

In this revised contribution the authors report on the generation and characterization of XUV and electron pulses carrying orbital angular momentum (OAM). The authors did a good job to revise the manuscript and answer my open questions. Among other revisions, they have provided additional evidence, that the observed ring shaped light beam is indeed an optical vortex with the claimed orbital angular momentum. In my opinion the paper can be accepted now, after some additional revisions:

1) In the title light and electronic vortices are claimed. As mentioned above, the authors have provided enough evidence, that they have generated optical vortices at short wavelengths. However, only very little evidence is provided for the generation of attosecond electronic vortices. The claim is based only on the physics of photoionization, which is probably true, but such a major claim needs more evidence or at least discussion. Remember, based on calculations the existence of attosecond light pulses have been also claimed for a long time, but it took some time and efforts for the first experimental proof. So weaken the claim and use a less general title, and/or provide at least a discussion how these attosecond electronic vortices can be experimentally verified.

2) Minor change, please reference all figures including the panels in the text (e.g Fig 3c and 3d are not mentioned in the text)

3) In line 124-127 the authors have described their claim of an OAM of 57. Please make also a similar revision in lines 304-305.

4) Supplementary information section 4: The discussion about the evolution of the beam diameter is very interesting and helpful. I have one minor question: the initial beam diameter is assumed to be 60 μm . Will it be the same for all harmonics? Due to the highly nonlinear nature of HHG, it is expected, that higher order harmonics are mainly generated close to the axis, i.e. the beam diameter depends on the harmonic order. Please add a comment, how this will influence your findings.

Reviewer #2 (Remarks to the Author):

The authors have responded my concerns and the quality of the manuscript has improved. The point that I have found more relevant is that they have given more impact to their second measurement in contrast to the first one. For instance, concerning the RABBITT measurement of the attosecond XUV vortices, I found very positive the inclusion of Figure 5 in the Supplemental Material, that unequivocally demonstrates the validity of the results presented in Figure 3 of the main text. I recommend the manuscript for publication in Nature Communications.

I have some minor comments that do not compromise my decision of recommending the manuscript for publication:

1. Regarding one of my previous comments, the authors have included in lines 100-101: "The scaling of the divergence is however very different from the usual Gaussian case, for which the harmonic divergence noticeably changes with harmonic order." Although this result is well-known in the HHG community, the inclusion of some references would help the reader.

2. Concerning the same comment about figure 1, and my previous comment number 6, I found disappointing that the authors didn't include the direct comparison between theoretical and experimental results in Fig. 1. If comparing the ring diameters of Figs. 1 and 2, they do not agree, and the authors should comment on it. In contrast, regarding Fig. 3, although theoretical calculations will make their results clearer, the inclusion of experimental Fig. 5 in the Supplemental Material unequivocally demonstrates the validity of the method for characterizing XUV attosecond vortices.

Reviewer #3 (Remarks to the Author):

This is a re-submission of the paper "Attosecond light and electronic vortices".

In my first review I suggested that the original manuscript would be greatly improved by removing the artificial controversy over the conservation of orbital angular momentum as a justification for the paper. I am pleased to see that this has been done in the revised manuscript. Conservation of orbital angular momentum is a consequence of phase matching for a thin medium such as the authors use. No paper is improved by hiding a clear interpretation in seeming complexity.

The electron vortices is a similar case. Photoionization transfers amplitude and phase information from photons to electrons. If the light field has orbital angular momentum, so does any electron created by it.

In my opinion, the RABBIT measurement is a significant contribution to how to characterize orbital angular momentum beams created by high harmonic generation and warrants publication.

Reviewer #1 (Remarks to the Author):

In this revised contribution the authors report on the generation and characterization of XUV and electron pulses carrying orbital angular momentum (OAM). The authors did a good job to revise the manuscript and answer my open questions. Among other revisions, they have provided additional evidence, that the observed ring shaped light beam is indeed an optical vortex with the claimed orbital angular momentum. In my opinion the paper can be accepted now, after some additional revisions:

1) In the title light and electronic vortices are claimed. As mentioned above, the authors have provided enough evidence, that they have generated optical vortices at short wavelengths. However, only very little evidence is provided for the generation of attosecond electronic vortices. The claim is based only on the physics of photoionization, which is probably true, but such a major claim needs more evidence or at least discussion. Remember, based on calculations the existence of attosecond light pulses have been also claimed for a long time, but it took some time and efforts for the first experimental proof. So weaken the claim and use a less general title, and/or provide at least a discussion how these attosecond electronic vortices can be experimentally verified.

We agree with this statement. The title of the article has been changed to "Synthesis and characterization of attosecond light vortices in the extreme ultraviolet" to better reflect its contents.

2) Minor change, please reference all figures including the panels in the text (e.g Fig 3c and 3d are not mentioned in the text)

Fig 3c and 3d are now respectively referenced at lines 264 and 268.

3) In line 124-127 the authors have described their claim of an OAM of 57. Please make also a similar revision in lines 304-305.

Line 370 has been changed to "*The maximum value of OAM obtained was $\ell_{19} = 57$ using $\ell_1 = 3$ in argon.*"

4) Supplementary information section 4: The discussion about the evolution of the beam diameter is very interesting and helpful. I have one minor question: the initial beam diameter is assumed to be 60 μm . Will it be the same for all harmonics? Due to the highly nonlinear nature of HHG, it is expected, that higher order harmonics are mainly generated close to the axis, i.e. the beam diameter depends on the harmonic order. Please add a comment, how this will influence your findings.

Indeed for a Gaussian driver the higher orders come from the higher intensity region which is $R=0$. However in the case of a Laguerre-Gaussian driver, the higher intensity region is found along the ring;

more precisely, at $R_{\text{max}} = w_0 \sqrt{\frac{|\ell_1|}{2}}$ as explained in Supplementary Note 1. Therefore the beam diameter (i.e. the diameter of the ring) stays the same for all harmonic orders, as depicted in the picture below (which is not the result of a calculation and only serves as an illustration).

Reviewer #2 (Remarks to the Author):

The authors have responded my concerns and the quality of the manuscript has improved. The point that I have found more relevant is that they have given more impact to their second measurement in contrast to the first one. For instance, concerning the RABBIT measurement of the attosecond XUV vortices, I found very positive the inclusion of Figure 5 in the Supplemental Material, that unequivocally demonstrates the validity of the results presented in Figure 3 of the main text. I recommend the manuscript for publication in Nature Communications.

I have some minor comments that do not compromise my decision of recommending the manuscript for publication:

1. Regarding one of my previous comments, the authors have included in lines 100-101: "The scaling of the divergence is however very different from the usual Gaussian case, for which the harmonic divergence noticeably changes with harmonic order." Although this result is well-known in the HHG community, the inclusion of some references would help the reader.

A reference to "He, X. et al. Spatial and spectral properties of the high-order harmonic emission in argon for seeding applications. Phys. Rev. A 79, 063829 (2009)." was added at line 124.

2. Concerning the same comment about figure 1, and my previous comment number 6, I found disappointing that the authors didn't include the direct comparison between theoretical and experimental results in Fig. 1. If comparing the ring diameters of Figs. 1 and 2, they do not agree, and the authors should comment on it. In contrast, regarding Fig. 3, although theoretical calculations will make their results clearer, the inclusion of experimental Fig. 5 in the Supplemental Material unequivocally demonstrates the validity of the method for characterizing XUV attosecond vortices.

It would have been indeed satisfying to compare those two quantities. Unfortunately, there is too much uncertainty in the experimental data about the determination of the divergence: the focus position as well as the observation distance are not very well known. Moreover, the generation intensity is difficult to estimate experimentally and influences directly the harmonic divergence through the atomic phase. All these unknown parameters are required to properly compare the theory and the experiment, which prevents us from doing the comparison.

However, we can still comment on the diameter ratios between $\ell=1, 2, 3$ in both theory and experiment, and as noted at lines 163-164, these ratios agree within 5%.

Reviewer #3 (Remarks to the Author):

This is a re-submission of the paper "Attosecond light and electronic vortices".

In my first review I suggested that the original manuscript would be greatly improved by removing the artificial controversy over the conservation of orbital angular momentum as a justification for the paper. I am pleased to see that this has been done in the revised manuscript. Conservation of orbital angular momentum is a consequence of phase matching for a thin medium such as the authors use. No paper is improved by hiding a clear interpretation in seeming complexity.

The electron vortices is a similar case. Photoionization transfers amplitude and phase information from photons to electrons. If the light field has orbital angular momentum, so does any electron created by it.

In my opinion, the RABBIT measurement is a significant contribution to how to characterize orbital angular momentum beams created by high harmonic generation and warrants publication.